# The Effect of Prolactin on Gene Expression and the Secretion of Reproductive Hormones in Ewes during the Estrus Cycle

**DOI:** 10.3390/ani14131873

**Published:** 2024-06-25

**Authors:** Sicong Yue, Jiaxin Chen, Chunhui Duan, Xiangyun Li, Ruochen Yang, Meijing Chen, Yu Li, Zhipan Song, Yingjie Zhang, Yueqin Liu

**Affiliations:** College of Animal Science and Technology, Hebei Agricultural University, Baoding 071000, China; yueyueyuexiaojuan@163.com (S.Y.); chenjiaxin1226@163.com (J.C.); duanchh211@126.com (C.D.); lxyun@hebau.edu.cn (X.L.); yangruochen1110@126.com (R.Y.); chenmeijing815@126.com (M.C.); liyu18730280787@163.com (Y.L.); 18833468223@163.com (Z.S.)

**Keywords:** prolactin, ovary, serum hormone, reproduction, sheep

## Abstract

**Simple Summary:**

Prolactin (PRL) plays an important role in animal molecule development and ovulation. However, the regulatory effects on the different stages of the estrus cycle in ewes are unclear. We studied the effect of PRL on the secretion of reproductive hormones, follicle counts, and gene expressions via the PRL inhibitor bromocriptine. Our findings show that PRL had no significant effect on the of the estrus cycle. PRL inhibition affected the serum concentrations of E_2_, FSH, and GnRH, as well as the expression of PR, FSHR, LHR, 3β-HSD, StAR, CYP11A1, CYP19A1, Bax, Bcl-2, and Caspase-3 in different stages of the estrus cycle. These results provide a basis for understanding the mechanisms underlying the regulation of the ewe estrus cycle by PRL.

**Abstract:**

Prolactin (PRL) plays an important role in animal follicle development and ovulation. However, its regulatory effects on the different stages of the estrus cycle in ewes are unclear. In this study, bromocriptine (BCR, PRL inhibitor) was used to study the effect of PRL on the secretion of reproductive hormones and gene expressions in order to explore its regulatory effects on the sexual cycle of ewes. Eighty healthy ewes with the same parity and similar weights were randomly assigned to the control group (C, n = 40) and the treatment group (T, n = 40, fed bromocriptine). After estrus synchronization, thirty-one ewes with overt signs of estrus were selected from each group. Six blood samples were randomly obtained by jugular venipuncture to measure the concentration of PRL, estrogen (E_2_), progesterone (P_4_), luteinizing hormone (LH), follicle-stimulating hormone (FSH), and gonadotropin-releasing hormone (GnRH) in the proestrus, estrus, metestrus, and diestrus. At the same time, we collected the ovaries of the six ewes in vivo after anesthesia in order to detect follicle and corpus luteum (CL) counts and measure the expression of hormone-receptor and apoptosis-related genes. The results show that PRL inhibition had no significant effects on the length of the estrus cycle (*p* > 0.05). In proestrus, the number of large and small follicles, the levels of E_2_, FSH, and GnRH, and the expressions of *ER*, *FSHR*, and the apoptotic gene *Caspase-3* were increased (*p* < 0.05); and the number of middle follicles and the expression of anti-apoptotic gene *Bcl-2* were decreased (*p* < 0.05) in the T group. In estrus, the number of large follicles, the levels of E_2_ and GnRH, and the expressions of the *StAR*, *CYP19A1*, and *Bcl-2* genes were increased (*p* < 0.05), and the number of middle follicles was decreased (*p* < 0.05) in the T group. In metestrus, the number of small follicles and the expression of *LHR* (*p* < 0.05) and the pro-apoptotic gene *Bax* were increased (*p* < 0.05); the number of middle follicles was decreased (*p* < 0.05) in the T group. In diestrus, the number of large follicles, middle follicles, and CL, the level of P_4_, and the expressions of *PR*, *3β-HSD*, *StAR*, *Caspase-3*, and *Bax* were increased (*p* < 0.05); the number of small follicles and the expression of *Bcl-2* were decreased (*p* < 0.05) in the T group. In summary, PRL inhibition can affect the secretion of reproductive hormones, the follicle count, and the gene expression during the estrus cycle. These results provide a basis for understanding the mechanisms underlying the regulation of the ewe estrus cycle by PRL.

## 1. Introduction

An ewe’s estrus cycle is 16–17 days [1]. It consists of four stages, including the proestrus, estrus, metestrus, and diestrus [2], which involve the activation and growth of primordial follicles, the selection and maturation of the dominant follicle, and ovulation [3]. Hormones regulate follicular development and atresia [4]. Follicle development is divided into the follicle-stimulating hormone (FSH) and luteinizing hormone (LH)-dependent stages; a decrease in hormone secretion leads directly to follicular atresia [5]. LH and FSH act directly on the theca externa and the GCs of follicles, inducing the final and nuclear maturation stages of oocytes, as well as the subsequent rupture of follicles and the formation of corpus luteum [6]. In GCs, estradiol 17 beta (E_2_) assumes a central role in follicular development and selection by activating estrogen receptors beta (*ER*) [7]. The intrafollicular P_4_ concentration is influenced by the presence of the corpus luteum (CL) and modulates the biological processes related to follicular cell development and oocyte competence [8]. In sheep follicle development and ovulation, endocrine regulation can greatly improve the reproductive rate and economic benefits of sheep.

Prolactin (PRL) is primarily secreted by the lactotrophs of the pituitary, acting via the prolactin receptor (*PRLR*) on the target cell [9]. In addition to regulating lactation [10], growth performance [11], animal behavior [12], and metabolism [13], PRL also plays a crucial role in reproductive processes such as follicle development and ovulation, with changes in its concentration particularly closely related to estrus and ovulation [14,15]. Studies have shown that PRL has important modulatory effects on the reproductive system in sheep via inhibitory actions on pituitary gonadotrophs and hypothalamic gonadotrophin hormone release [16], regulating the production of reproductive hormones including E_2_ [17], P_4_ [18], LH [19], FSH [20], and GnRH [21]. A reduction in PRL indirectly leads to an increase in the LH pulse frequency, which regulates follicular development [22]. Blaszczyk et al. showed that the PRL concentrations of Anglo-Nubian dairy goats in Poland were significantly higher during the non-breeding season than during the breeding season [23]. Reducing PRL during non-breeding seasons induces off-season estrus in sheep [24]. Bromocriptine (BCR), a commonly used dopamine agonist [25,26,27], can reduce PRL levels in the body [28], thus alleviating hypogonadism, infertility, galactorrhea, oligomenorrhea, and amenorrhea due to serum PRL elevation [29,30]. However, the regulation of the estrus cycle in sheep via PRL inhibition remains unclear.

The aim of this study was to determine whether PRL plays a regulatory role in the sexual cycle of ewes. In this study, bromocriptine (BCR, PRL inhibitor) was used to investigate the effects of PRL inhibition on serum reproductive hormones and reproduction-related genes in ewes during the proestrus, estrus, metestrus, and diestrus stages.

## 2. Materials and Methods

### 2.1. Animals and Feeding Management

This study was conducted during the breeding season (November) at the Zhihao Livestock Science and Technology Corporation in Wuyi, Hengshui, China. All procedures utilized in this study were approved by the Laboratory Animal Ethics Committee at Hebei Agricultural University (Hebei, China; permit number 2023156). All ewes had free access to fresh water and were fed twice daily (07:00 and 15:00 h) throughout the experiment. All ewes were housed in individual pens. Ewes were fed a basal diet, as shown in Appendix A.

### 2.2. Experimental Design

#### 2.2.1. Sheep Estrus Synchronization Treatment of Ewes

Eighty healthy non-pregnant ewes (Hu sheep, 2–3 years, body weight = 52.98 ± 0.96 kg) were selected and randomly divided into a control group (C) and a treatment group (T). The estrus synchronization protocol consisted of the insertion of an implanted progesterone sponge plug on Day 0 (MAP, 45 mg/piece, SYNCRITE-45 Vaginal Sponge, Australia), the removal of the device at 16:00 on Day 11, and an injection of 330 IU equine chorionic gonadotrophin (eCG; Sansheng Pharmaceutical Ltd., Ningbo, China) on Day 11. Ewes were checked for estrus (acceptance of male) with a vasectomized buck every four hours during presumed estrous periods and twice daily during other periods. After estrus synchronization, thirty-one ewes with overt signs of estrus were selected randomly from each group. The experiment lasted 38 days.

#### 2.2.2. Experimental Design

A schematic representation of the main activities performed during the whole experimental period is depicted in Figure 1. We began feeding the ewes in the T group with PRL inhibitor (bromocriptine (BCR), 2.5 mg/d, dissolved in water and evenly sprayed in the feed) at 0 d. The estrus was checked again 14 days after being induced, and ewes were on spontaneous estrus. Ewes in proestrus were determined according to records of estrus cycles: the proestrus stage is 1 day before the estrus, the estrus stage is 1 day after the proestrus, the metestrus stage is 2 days after estrus, and the diestrus stage is 7 days before the estrus. The first and second estrus times were accurately recorded, and the number of estrus cycle days was counted by subtracting the first estrus date. In the proestrus (I), estrus (II), metestrus (III), and diestrus (IV) stages, six ewes per stage were randomly selected for blood and ovarian tissue collection form both the C (n = 6) and T (n = 6) groups. The number of follicles with a diameter of ≥ 1 mm was observed and recorded. Those sheep whose ovaries were collected were no longer involved in the experiment, while we continued to feed the rest BCR.

### 2.3. Blood and Ovary Collection and of Follicle Count Statistics

Before the morning feeding, blood samples from six randomly selected sheep in the C and T groups were collected via jugular venipuncture into 5 mL coagulation-promoting tubes during the proestrus (I), estrus (II), metestrus (III), and diestrus (IV) stages. The samples were immediately centrifuged at 3000× *g* for 15 min to harvest serum and stored at −20 °C until analysis. Ovarian tissue was surgically collected, and the numbers of follicles and corpus luteum were recorded, for which 1–2 mm was considered small follicles, 2–4 mm medium follicles, and >4 mm large follicles [31,32,33]. All surgeries were performed under sodium pentobarbital anesthesia with efforts made to minimize animal suffering.

After rinsing the ovarian tissue with RNase-free phosphate-buffered saline (PBS), it was cut into 1 cm^3^ pieces in a sterile environment before immediately being frozen in liquid nitrogen and stored in a refrigerator at −80 °C for subsequent RNA extraction.

### 2.4. Reproductive Hormone Assays

Following the manufacturer’s instructions, commercial sheep enzyme-linked immunosorbent assay (ELISA) kits from Nanjing Jiancheng Bio, Nanjing, China, were used to determine the serum concentrations of PRL (H095-1-2, sensitivity > 0.1 ng/mL), FSH (H101-1-2, sensitivity > 0.1 mIU/mL), E_2_ (H102-1-2, sensitivity > 0.1 ng/L), LH (H206-1-2, sensitivity > 0.1 mIU/mL), progesterone (P_4_, H089-1-1, sensitivity > 0.1 ng/mL), and gonadotropin-releasing hormone (GnRH, H297, sensitivity > 0.1 ng/L). The absorbance (OD) of each well was measured at 450 nm and a standard curve was generated. According to the standard curve, the serum hormone concentration of the test sheep was calculated. The intraassay CV was 10%.

### 2.5. Determination of Relative Gene Expression

Total RNA was extracted using the RNAprep Pure Tissue Kit (TIANGEN, DP431, Beijing, China) and stored at −80 °C. cDNA was synthesized using the HiFiScript gDNA Removal RT MasterMix from Cowin Biotech Co., Ltd. (CWBIO, Jiangsu, China). The protocol was followed according to the manufacturer’s instructions.

The primers (Table 1) were created using Primer Premier 5.0 software and synthesized using BGI·Write (Beijing, China). The TransStart^®^ Tip Green qPCR SuperMix (+Dye I) from TransGen Biotech, Beijing, China was used for qRT-PCR on an ABI QuantStudio 7 Flex System (Foster City, CA, USA). Each sample was tested in triplicate using quantitative real-time PCR with GAPDH as an endogenous reference gene. Relative expression levels were calculated using the 2^−ΔΔCt^ method.

### 2.6. Statistical Analysis

The length of the estrus cycle, hormone levels, and relative expressions were compared among multiple groups using the one-way ANOVA procedure in SPSS software (ver. 22.0, IBM Corp., Armonk, NY, USA), followed by Duncan’s post hoc test. The results were expressed as mean ± standard error of the mean (SEM), and statistical significance was defined as *p* < 0.05. Visualization mapping was performed using GraphPad Prism 9.0 software.

## 3. Results

### 3.1. Effects of BCR on Serum Reproductive Hormone in Ewes at Different Stages of the Estrus Cycle

The effects of BCR on the length of the estrus cycle in ewes are shown in Figure 2A; there was no significant difference between the C and T groups (*p* > 0.05). The effects of BCR on reproductive hormones at each stage of the ewes’ estrus cycle are shown in Figure 2B–G. In the proestrus, estrus, metestrus, and diestrus stages of the estrus cycle, BCR significantly decreased the serum concentrations of PRL (*p* < 0.05). In the proestrus stage, the serum concentrations of E_2_, FSH, and GnRH in the T group were significantly increased (*p* < 0.05) and no significant differences occurred in P_4_ and LH (*p* > 0.05). In the estrus stage, the serum concentrations of E_2_ and GnRH in T the group were significantly increased (*p* < 0.05), and no significant differences occurred in the contents of P_4_, LH, and FSH (*p* > 0.05). In the metestrus stage, there was no significant difference in other reproductive hormones, except PRL, between the two groups (*p* > 0.05). In the diestrus stage, the serum concentration of P_4_ in the T group was significantly increased (*p* < 0.05), while no significant difference occurred in E_2_, LH, FSH, and GnRH (*p* > 0.05).

### 3.2. Effects of BCR on the Number of Follicles and Corpus Luteum in Ewes at Each Stage of the Estrus Cycle

The changes in the number of large, medium, and small follicles and the number of CLs in the ewes’ estrus cycle are shown in Figure 3. In the proestrus stage, the number of large and small follicles in the T group was increased (*p* < 0.05), while the number of middle follicles was decreased (*p* < 0.05). In the estrus stage, the number of large follicles was increased (*p* < 0.05), the number of middle follicles was decreased (*p* < 0.05), and the number of small follicles in the T group was not significantly different (*p* > 0.05). In the metestrus stage, the number of small follicles was increased (*p* < 0.05), the number of middle follicles was decreased (*p* < 0.05), and the number of large follicles in the T group was not significantly different (*p* > 0.05). In the diestrus stage, the number of large follicles, middle follicles, and CLs was increased (*p* < 0.05), while the number of small follicles was decreased (*p* < 0.05).

### 3.3. Effects of BCR on mRNA Expression of Genes at Each Stage of the Estrus Cycle in Ewes

The mRNA expression of the hormone-receptor genes during the ewes’ estrus cycle is shown in Figure 4. The expressions of *L-PRLR* and *S-PRLR* were significantly different during the estrus cycle (*p* < 0.05). After PRL inhibition, the expression level of *L-PRLR* increased in the proestrus and decreased in the estrus, metestrus, and diestrus stages (*p* < 0.05). The expression of *S-PRLR* decreased in the metestrus and increased in the proestrus, estrus, and diestrus stages (*p* < 0.05). In the proestrus stage, the expressions of *FSHR* and *ER* in T group were increased (*p* < 0.05) and *PR* was decreased (*p* < 0.05). In the estrus stage, the expression of *ER* was increased (*p* < 0.05) and the expressions of *FSHR*, *LHR*, and *PR* were decreased (*p* < 0.05). In the metestrus stage, the expressions of *FSHR*, *ER*, and *LHR* in the T group were increased (*p* < 0.05), and there was no significant difference in the expression of *PR* (*p* > 0.05). In the diestrus, the expressions of *LHR* and *PR* in the T group were increased (*p* < 0.05), while the expression of *FSHR* was decreased (*p* < 0.05) and no significant difference occurred in the expression of *ER* (*p* > 0.05).

The expressions of steroidogenic enzymes are shown in Figure 5. In the proestrus stage, the expression of *CYP19A1* in the T group was increased (*p* < 0.05), while the expression of *CYP11A1* was decreased (*p* < 0.05). In the estrus stage, the expressions of *StAR* and *CYP19A1* were increased. In the metestrus stage, the expressions of *StAR* and *CYP11A1* in the T group were increased (*p* < 0.05), and the expressions of *3β-HSD* and *CYP19A1* were decreased. In the diestrus stage, the expressions of *3β-HSD* and *StAR* were increased (*p* < 0.05), while the expressions of *CYP19A1* and *CYP11A1* were not significantly different (*p* > 0.05).

The expressions of apoptosis-related genes are shown in Figure 6. In the proestrus stage, the expression of *Caspase-3* was increased (*p* < 0.05) while that of *Bcl-2* was decreased (*p* < 0.05) in the T group. In the estrus stage, the expression of *Bcl-2* was increased (*p* < 0.05), while there was no significant difference in the expressions of *Caspase-3* and *Bax* (*p* > 0.05). In the metestrus stage, the expression of *Bax* was increased in the T group (*p* < 0.05), but there was no significant difference in *Bcl-2* and *Caspase-3* (*p* > 0.05). In the diestrus stage, the expressions of *Caspase-3* and *Bax* in the T group were increased (*p* < 0.05), while the expression of *Bcl-2* was decreased (*p* < 0.05).

## 4. Discussion

### 4.1. Effect of PRL on Follicle Count and CL Number

In dairy cows, suckling elevated levels of PRL lead to anestrus in dairy cows [34]. PRL levels are correlated with the duration of postpartum amenorrhea [35]; in rats, these levels have been shown to decline postpartum after separation from the pup [36]. During the estrus cycle, the number of follicles varied dynamically with the follicle wave [37]. The number and size of follicles were somewhat indicative of ovarian activity [38]. High levels of PRL significantly inhibited the diameter and number of follicles in rats [39]; these results align with those in our study, that the inhibition of PRL significantly increased the number of large follicles in the proestrus, estrus, and metestrus stages. In addition, this study showed that PRL inhibition increased the proportion of follicles developing into large follicles. This is consistent with the results of Picazo’s study examining the 2–3 mm medium follicle count (*p* < 0.01) in the ovaries of Spanish merino ewes, as detected in the follicular phase after BCR injection [26]. We also found that the number of follicles increased in the diestrus stage after BCR feeding; this was associated with high levels of FSH in the proestrus, producing more follicles [40]. The CL is a temporary endocrine gland formed during ovulation [41]. The addition of PRL inhibited hCG-induced ovulation in a dose-related fashion [42]. We found that PRL inhibition significantly increased the number of CLs in the diestrus stage, which is consistent with similar histopathological results from rat ovaries [43]. Thus, this suggests that PRL inhibition can promote follicle development and ovulation during the estrus cycle.

### 4.2. Effect of PRL on the Secretion of Related Reproductive Hormones

Endocrine pathways play an essential role in the menstrual cycle [44], throughout which hormone levels fluctuate [45], regulating estrus production [46] and ovulation [47]. Our study showed that PRL inhibition significantly increased the concentrations of GnRH, FSH, and E_2_ as well as the number of small follicles in the proestrus stage and increased the concentrations of GnRH and E_2_ in the estrus stage. A high PRL concentration suppresses the hypothalamic gonadotropin-releasing hormone, which plays a pivotal role in reproduction by stimulating the synthesis and secretion of LH and FSH [47]. Upon FSH stimulation, a cohort of small antral follicles begins the gonadotropin-dependent development phase of recruited follicles [48]. With the growth of recruited follicles, more and more E_2_ is produced and FSH release is suppressed [1]. A higher level of PRL was also shown to significantly reduce E_2_ production in granule cells (GCs) [49]. Our study found that the P_4_ concentration was significantly increased after PRL inhibition in the diestrus stage, mediating the further regulation of estrus follicle development. P_4_ is mainly produced by the CL [50] and its concentration _44_ could be due to an increased number of CLs [51]. The inhibition of PRL increased the P_4_ concentration in heifer [52], which is consistent with our research; PRL has also been significantly negatively correlated with P_4_ levels during the luteal phase [53]. Therefore, PRL inhibition can regulate the production of recruiting follicles and ovulation by increasing GnRH and E_2_ in the proestrus and estrus stages and increasing P_4_ in the diestrus stage.

### 4.3. Effect of PRL on the Expression of Genes at Different Stages of the Estrus Cycle

PRL plays a crucial role in reproduction by binding with different PRLRs [54]. Ruminants have two types of PRLR, long PRLR (*L-PRLR*) and short PRLR (*S-PRLR*) [55], which serve different roles in the ovaries. Previous studies found that, while there is a marked increase in *L-PRLR* expression, the expression of *S-PRLR* remains constant during the estrus cycle in sheep [56]; this is in accordance with the fact that we examined sheep without BCR. During the cycle, we found that after PRL inhibition, the fluctuating expression of *S-PRLR* increased significantly in the proestrus, estrus, and diestrus stages. Thompson’s study suggests that *S-PRLR* is involved in the formation and maintenance of the CL [57], suggesting that the inhibition of PRL concentration may promote the formation and maintenance of the CL in the estrus cycle by regulating *S-PRLR*.

Our study showed that PRL inhibition could increase FSH sensitivity and further affect follicular development by increasing the levels of E_2_, FSH, *ER*, and *FSHR* in the proestrus stage. *ER* and *PR* are members of the nuclear receptor family [58], with transcription after combining with E_2_ [50], thus further regulating antral follicle development and oocyte maturation [59]. Physiologic levels of PRL amplify the stimulatory effects of FSH on the acquisition of *FSHR* production in cultured GCs, while higher concentrations of PRL cause a decrease in *FSHR* binding [60]. Only the follicles with the highest sensitivity to FSH, which are those follicles with the highest *FSHR* expression in GCs, can secure the dominance of the selected follicle [61]; conversely, follicles lead to GC apoptosis, and ultimately follicular atresia [62]. In the estrus stage, we found that PRL inhibition resulted in a significant increase in *StAR* and *CYP19A1*. *StAR* is an enzyme that controls the rate of cholesterol transport to the inner mitochondrial membrane [63], and the expression of the *StAR*, *CYP11A1,* and *CYP19A1* genes increases with follicle development [64], suggesting that the inhibition of PRL in the estrus stage can significantly promote cholesterol transport and follicle development. In the metestrus stage, we found a significant increase in *LHR* levels, but no significant difference occurred in LH levels. BCR or PRL treatments alone did not affect the LH response [65]. LH acted on mural GCs, yielding high levels of *LHR* and initiating a series of events leading to cumulus cell expansion and follicle rupture [66]. With the increase in *LHR* gene expression, the sensitivity of follicles to LH increased, which drove follicular development in the second half of the follicular phase [48,67]. In the present study, the suppression of PRL promoted the expression of the pro-apoptotic gene *Bax*. *Bax* is more predominantly found in atresia follicles [68]. In the metestrus stage, ovulation leads to an increase in the expression of pro-apoptotic genes [69]. This further confirms our results, in that PRL inhibition in the metestrus stage may be due to increased LH sensitivity, which then promotes follicle rupture and ovulation.

CL regression is mediated by apoptosis [70], which initiates luteolysis by promoting extrinsic apoptosis and destructive autophagy [71], thereby decreasing P_4_ secretion [72]. The early increase in plasma P_4_ concentration during the luteal phase promotes the premature activation of the luteolytic process, thus affecting CL function in llamas [73]. We found that both P_4_ and *PR* in the diestrus stage increased significantly after PRL inhibition; the expression of *3β-HSD*, *StAR*, *Bax*, and *Caspase-3* was increased, and the expression of *Bcl-2* was decreased, thus demonstrating that PRL inhibition could improve the diestrus steroidogenic ability and induce luteinization and apoptosis.

## 5. Conclusions

Form our findings, we can conclude that PRL had no significant effect on the length of the estrus cycle. PRL inhibition affected the serum concentrations of E_2_, FSH, and GnRH, as well as the expressions of *PR*, *FSHR*, *LHR*, *3β-HSD*, *StAR*, *CYP11A1*, *CYP19A1*, *Bax*, *Bcl-2,* and *Caspase-3* in different stages of the estrus cycle. These results provide a basis for understanding the mechanisms underlying estrus cycle regulation in ewes via PRL.

## Figures and Tables

**Figure 1 animals-14-01873-f001:**
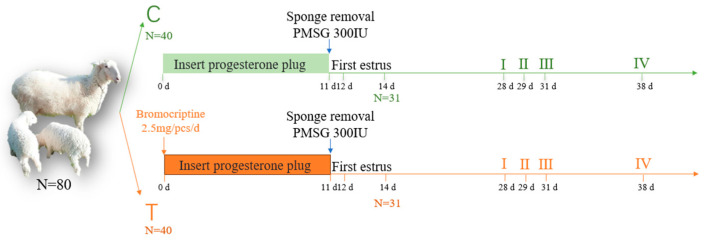
Schematic of the experimental design and collection protocol: a total of eighty sheep were randomly assigned to either the control (C) or treatment group (T). Blood and ovarian tissues were collected during the proestrus (I), estrus (II), metestrus (III), and diestrus (IV) stages of spontaneous estrus for both groups.

**Figure 2 animals-14-01873-f002:**
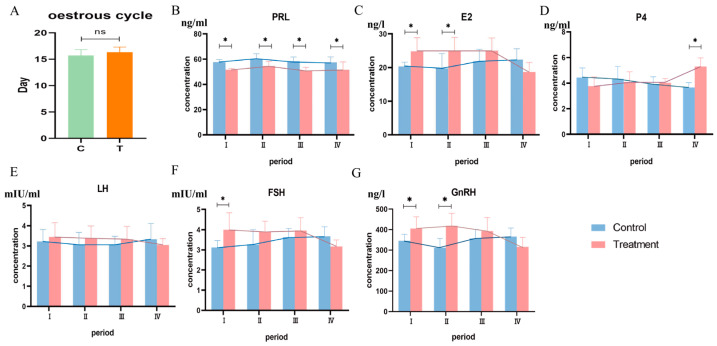
Effects of BCR on estrus cycle days and serum concentrations in ewes: (**A**) estrus cycle, (**B**) PRL, (**C**) E_2_, (**D**) P_4_, (**E**) LH, (**F**) FSH, (**G**) GnRH. ns: *p* > 0.05. * *p* < 0.05.

**Figure 3 animals-14-01873-f003:**
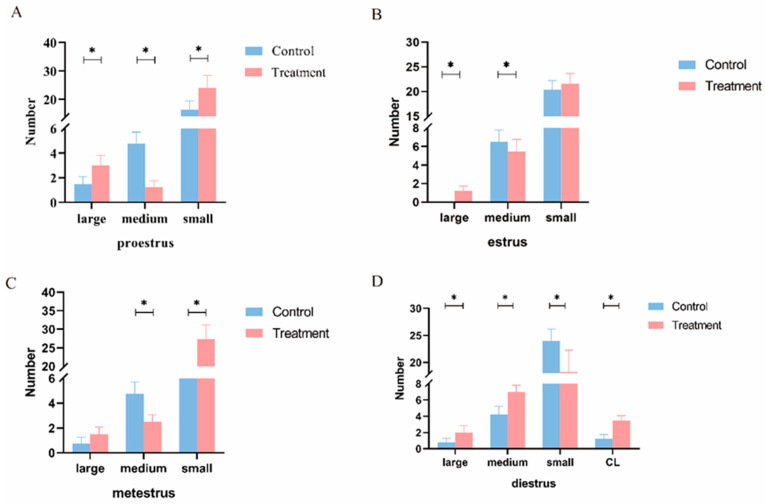
Effects of BCR on follicle numbers at different stages of the estrus cycle in ewes. (**A**) Number of follicles in proestrus, (**B**) number of follicles in estrus, (**C**) number of follicles in metestrus, (**D**) number of follicles in diestrus. Large: large follicle count; medium: middle follicle count; small: small follicle count; CL: corpus luteum. * *p* < 0.05. Six replicates per group.

**Figure 4 animals-14-01873-f004:**
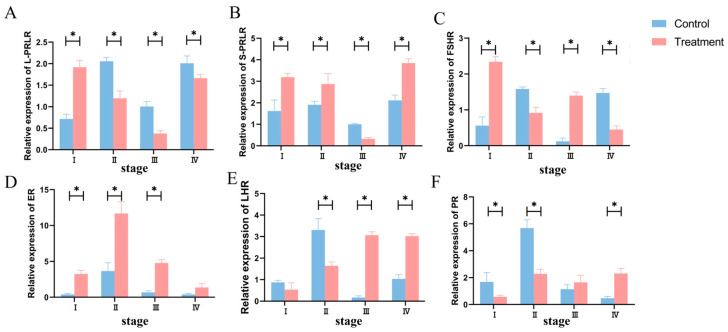
Effect of BCR on the mRNA expression of reproductive hormone-receptor (**A**–**F**) genes at different stages of the estrus cycle in ewes. (**A**) *L-PRLR*; (**B**) *S-PRLR*; (**C**) *FSHR*; (**D**) *ER*; (**E**) *LHR*; (**F**) *PR*. * *p* < 0.05.

**Figure 5 animals-14-01873-f005:**
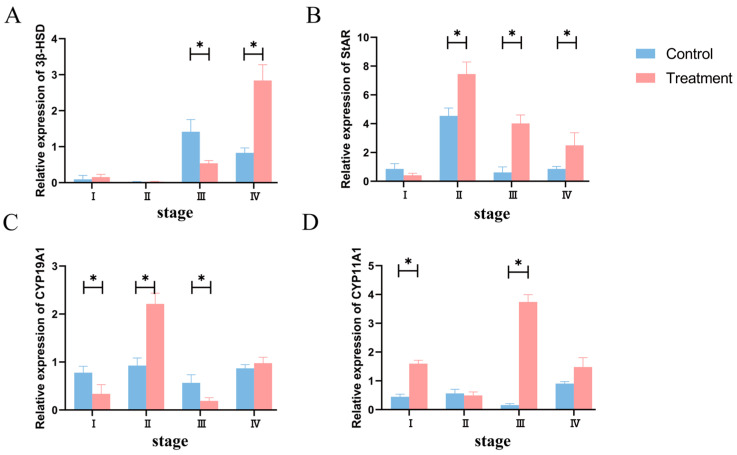
Effect of BCR on the mRNA expression of steroid-receptor (**A**–**D**) genes at different stages of the estrus cycle in ewes. (**A**) *3β-HSD*; (**B**) *StAR;* (**C**) *CYP19A1*; (**D**) *CYP11A1*. * *p* < 0.05.

**Figure 6 animals-14-01873-f006:**
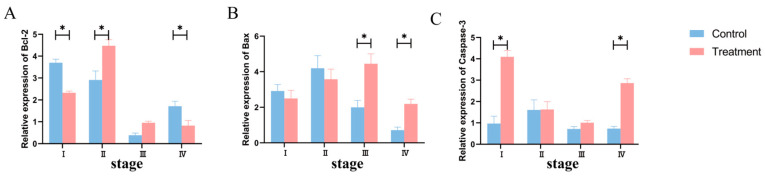
Effect of BCR on the mRNA expression of apoptosis-related (**A**–**C**) genes at different stages of the estrus cycle in ewes. (**A**) *Bcl-2*; (**B**) *Bax*; (**C**) *Caspase-3*. * *p* < 0.05.

**Table 1 animals-14-01873-t001:** List of primers used in qRT-PCR.

Gene	Sequence (5′-3′)	Size (bp)	Tm (°C)	Accession Number
*L-PRLR*	F: CCCCTTGTTCTCTGCTAAACCC R: CTATCCGTCACCCGAGACACC	129	60	O46561-1
*S-PRLR*	F:ACAGTAAGCGCCATCAACCA R: CTGGCTTGCATCGAATCTGC	328	60	O46561-2
*FSHR*	F: GTGACACCAAGATAGCCAAGCG R: GGGTAGAACAGGACCAGGAGGA	151	60	NM_001009289.1
*LHR*	F: ATCCAGAGCTGATGGCTACC	115	60	NM_001278566.2
	R: GCAGCTGAGATGGCAAAGAA			
PR	F: CAACAGCAAACCTGATACCT R: CCATCCTAGTCCAAATACCATT	183	60	XM_015100878.2
*ER*	F: CGGCTACGCAAGTGCTATGAA	385	60	XM_027972563.1
	R: CCACAAATCCTGGCACCCT			
*StAR*	F: ATTCAGGAGGCAAAGAGCAGC	270	60	XM_015094520.2
	R: TCGGGTAAGGAAAATGGGTCA			
*3β-HSD*	F: CAGTCTATGTTGGCAATGTGGC	283	60	NM_001135932.1
	R: CGGTTGAAGCAGGGGTGGTAT			
*CYP11A1*	F: GTTTCGCTTTGCCTTTGAGTC	120	60	NM_001093789.1
	R: ACAGTTCTGGAGGGAGGTTGA			
*CYP19A1*	F: GCTTTTGGAAGTGCTGAACCC	379	60	NM_001123000.1
	R: CATGCCGATGAACTGCAACC			
*Caspase-3*	F: AATGCAAGAAGCAGGGCACCCA	275	60	XM_015104559.3
	R:GGGTTACAGCGATGCAGAAGGTTCA			
*Bcl-2*	F:CGCTGAAGCGAAGCTGTAGA	176	60	XM_027960877.2
	R: CGTTGAGCCTGAAAGCTGTTT			
*Bax*	F: TGCCAGCAAACTGGTGCTCAA	183	60	XM_027978592.1
	R: GCACTCCAGCCACAAAGATGGT			
GAPDH	F: CTGACCTGCCGCCTGGAGAAA	149	60	NM001190390.1
	R: GTAGAAGAGTGAGTGTCGCTGTT			

## Data Availability

Data are contained within the article.

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
