# Peer review of "The Effect of Prolactin on Gene Expression and the Secretion of Reproductive Hormones in Ewes during the Estrus Cycle"

_animals, 2024, doi:10.3390/ani14131873_

Round 1

Reviewer 1 Report

Comments and Suggestions for Authors

This study attempts to explain the Effect of prolactin on secretion of reproductive hormones and gene expression of ewes during the estrus cycle”. The manuscript is interesting and within the scope of the journal. In some sections of the manuscript, the information provided is unclear.

I listed my other concerns below in the order I found them in the manuscript.

Simple Summary

The simple summary is not easy to follow. Please edit it.

Abstract

The abstract needs to be edited. The information presented is not easy to follow, and whether the authors describe the treatment group is unclear.

-        L18 – L19. Previous information has been published about this. This raises the question about the novelty of the study.

o   https://doi.org/10.1530/reprod/120.1.177

o   https://doi.org/10.1530/rep.1.00343

-        L23 – L24. It is not clear what “good estrus” means. Then, this selection was not made “randomly.” Was this selection equitable?

-        L24 – L26. It is not clear how this was done. Ovary samples? Reproductive hormones? Serum? Were the animals euthanized and the ovaries collected? How were the blood samples collected? Which hormones were analyzed?

-        L27 – L28. E2? LH? FSH?

-        L27 – L36. It is unclear whether the authors are describing the treatment group.

-        L36 – L38. These conclusions are not easy to follow.

Introduction

The Introduction is well-written and leads the reader to the objectives of the manuscript. However, the use of bromocriptine is not entirely justified. More information about this inhibitor is needed. What is the main role of this inhibitor? Is there a secondary effect of this inhibitor? Has it been used before (https://doi.org/10.1530/reprod/120.1.177)?

-        L56 – L57. I do not think this is the best reference to support this sentence. Is there a reference to cattle or goats that can support this?

-        L57 – L59. Please provide the breed and locations of this experiment.

-        L60 – L61. https://doi.org/10.1530/rep.1.00343

-        L62 – L65. The relationship between PRL and reproductive hormones is not entirely justified. Please add more information about the possible relationship among these hormones.

M&M

.

-        L72 – L76. The information is unclear. What are the requirements for raising and managing farm ewes? 62 selected? 62 from each group? Which groups? Basal diet? Based on what? Apporting what? Age and breed of the animals? Were these females sired by the same male? Were they coming from different farms? More information about the diet provided and the animals used is needed.

-         L79 – L80. 80? Same birth? Similar weight (BCS)? Control and treatment groups are unclear.

-        L86 – L87. Define “good estrus”. This number does not align with the previous number of selected animals.

-        L90. PC meaning? (2.5mg/pc/d)

-        L89 – L91. Is there any evidence that the water was not evaporated? Was the feed tested for the content of bromocriptine? The sentence is not clear (on the day of the burial of the embolism in the same estrus treatment)

-        L92 – L93. Please describe the process. How were the days of the estrus cycle counted?

-        L94 – L97. This number is changing over and over again. Please describe in detail the process. “Typical symptoms? How was the ovarian tissue collected? Blood?

-        L104 – L106. Randomly selected but with “good estrus”? 6 animals per stage and treatment?

-        L110 – L111. Who performed the surgeries?

-        L116 – L120. Melatonin? This hormone was not justified in the Introduction.

-        L137 – L142. More details about fixed effects are needed. Why was weight not used as a co-variant?

Results

Well explained.

Discussion

-        L219 – L222. This justification belongs to the Introduction.

-        L232 – L235. Number of CL? More than one? Why did the CL number increase?

-        L235 – L236. Positive or negative?

-        L250 GCs?

Conclusions

Please edit the conclusions.

-        L304 – L307. “PRL had no significant effect on…” “PRL affected…” It seems that PRL was injected.

Figures

-        Figure 2. Concentration of what? This applied to all the figures, except A.

-        Figure 3. Indicate the n per figure.

-        Figure 4 - 6. Title in X axis is missing.

Comments on the Quality of English Language

In some sections/sentences the English is unclear or with grammatical errors.

Author Response

Comments 1: The simple summary is not easy to follow. Please edit it.

Response 1: Thank you for pointing this out. We have added the corresponding information. We have revised part of the describe of simple summary, see the red section of the article for details.

Comments 2: The abstract needs to be edited. The information presented is not easy to follow, and whether the authors describe the treatment group is unclear.

-        L18 – L19. Previous information has been published about this. This raises the question about the novelty of the study.

o   https://doi.org/10.1530/reprod/120.1.177

o   https://doi.org/10.1530/rep.1.00343

-        L23 – L24. It is not clear what “good estrus” means. Then, this selection was not made “randomly.” Was this selection equitable?

-        L24 – L26. It is not clear how this was done. Ovary samples? Reproductive hormones? Serum? Were the animals euthanized and the ovaries collected? How were the blood samples collected? Which hormones were analyzed?

-        L27 – L28. E2? LH? FSH?

-        L27 – L36. It is unclear whether the authors are describing the treatment group.

-        L36 – L38. These conclusions are not easy to follow.

Response 2: Thank you for pointing this out. We have added the corresponding information. We revised the part of treatment group in the abstract, see the red section of the article for details.

L18-L19. These two articles are very related to our study, and we have also used both articles for reference. But we think the first study examined follicular and hormonal cycle changes in Spanish Merino sheep aged 5-6 years after bromohintine injection, missing the role of prolactin on hormone receptor genes and apoptosis genes during the cycle. And increasing age can lead to an increase in prolactin concentrations. The second study examined to determine the cellular localization and the changes in expression of long and short PRLr isoforms in sheep ovary throughout the estrous cycle. However, we further explored the expression of PRLRs by inhibiting PRL concentrations in sheep by feeding bromocriptine.

L23-24. We've changed the description of “good estrus”. What we want to express is that after the estrus synchronization, the sheep with high synchronization rate were selected, and the sheep with low synchronization rate were eliminated.

L24-26. We revised the relative description, see Line29-34 for details.

L27-28. E2, LH, FSH full name has been added.

L27 – L36. The description were changed.

We have revised part of the conclusion, see the red section of the article for details.

And we updated text in the manuscript and highlighted it.

Comments 3: The Introduction is well-written and leads the reader to the objectives of the manuscript. However, the use of bromocriptine is not entirely justified. More information about this inhibitor is needed. What is the main role of this inhibitor? Is there a secondary effect of this inhibitor? Has it been used before (https://doi.org/10.1530/reprod/120.1.177)?

-        L56 – L57. I do not think this is the best reference to support this sentence. Is there a reference to cattle or goats that can support this?

-        L57 – L59. Please provide the breed and locations of this experiment.

-        L60 – L61. https://doi.org/10.1530/rep.1.00343

-        L62 – L65. The relationship between PRL and reproductive hormones is not entirely justified. Please add more information about the possible relationship among these hormones.

Response 3:

Thank you for pointing this out. We agree with this comment. Therefore, we have added the corresponding information of bromocriptine in the manuscript.

L56 – L57. I have changed the new reference.

L57 – L59. the breed and locations were added in this experiment in line70.

L60 – L61. This reference mainly explores the localization and expression of PRLR in the cycle. However, studies on inhibition of PRL regulation in the cycle are unclear. We have changed our description in line86-88.

L62 – L65. I added reference in line 78 to justify the relationship among PRL and reproductive hormones.

Comments 4: M&M

L72 – L76. The information is unclear. What are the requirements for raising and managing farm ewes? 62 selected? 62 from each group? Which groups? Basal diet? Based on what? Apporting what? Age and breed of the animals? Were these females sired by the same male? Were they coming from different farms? More information about the diet provided and the animals used is needed.

-         L79 – L80. 80? Same birth? Similar weight (BCS)? Control and treatment groups are unclear.

-        L86 – L87. Define “good estrus”. This number does not align with the previous number of selected animals.

-        L90. PC meaning? (2.5mg/pc/d)

-        L89 – L91. Is there any evidence that the water was not evaporated? Was the feed tested for the content of bromocriptine? The sentence is not clear (on the day of the burial of the embolism in the same estrus treatment)

-        L92 – L93. Please describe the process. How were the days of the estrus cycle counted?

-        L94 – L97. This number is changing over and over again. Please describe in detail the process. “Typical symptoms? How was the ovarian tissue collected? Blood?

-        L104 – L106. Randomly selected but with “good estrus”? 6 animals per stage and treatment?

-        L110 – L111. Who performed the surgeries?

-        L116 – L120. Melatonin? This hormone was not justified in the Introduction.

-        L137 – L142. More details about fixed effects are needed. Why was weight not used as a co-variant?

Response 4:

L72 – L76.Thank you for pointing this out. We agree with this comment. Therefore, we have changed some information in the manuscript. Information about sheep was moved to the next part of the experiment design. And the composition of the basal diet were added in supplement table1.

L86 – L87. I have changed the description of good estrus. We want to mean that After estrus synchronization, 31 ewes were estrus response and then were choosed. Update in line 101-103.

 L90. PC meaning? (2.5mg/pc/d)--- I changed the description in line122

 L89 – L91. We have not consider the evaporation of water.  We considered the total amount of bromocriptine eaten by each sheep. And we changed the description of the sentence.

L92 – L93.  The estrus was checked again 14 days after being induced, and ewes were on spontaneous estrus. Ewes in proestrus were determined according to records of estrus cycles: the proestrus stage is 1 day before the estrus, the estrus stage is 1 day after the proestrus; the metes-trus stage is 2 days after estrus; and the diestrus stage is 7 days before the estrus. The first and second estrus times were accurately recorded, and the number of estrus cycle days was counted by subtracting the first estrus date.  The change were in line124-130.

L94 – L97. This number is changing over and over again. Please describe in detail the process. “Typical symptoms? How was the ovarian tissue collected? Blood?

After estrus synchronization, C group and T group were 31 ewes. At the second estrus, six ewes of  both C group and T group were collected per stage.  See the next section for specific acquisition steps.

L104 – L106. We changed the description of good estrus. And six ewes were collected in both C group and T group per stage.

L110 – L111. Who performed the surgeries? The surgery was performed by ourselves.

L116 – L120. Melatonin? It was our negligence that we removed this error from the manuscript.

L137 – L142. The weight of the ewes had been measured and selected before the start of the trial, and there was no significant difference in the weight of the ewes between the two groups.

Comments 5: Results

Well explained..

Response 5: Thank you very much.

Comments 6: Discussion

-        L219 – L222. This justification belongs to the Introduction.

-        L232 – L235. Number of CL? More than one? Why did the CL number increase?

-        L235 – L236. Positive or negative?

-        L250 GCs?

Response 6:

Thank you for pointing this out. We agree with this comment. Therefore, we have changed some information in the manuscript.

We've adjusted the position of the BCR description to introduction.

The number of CL are more than one. The details can be found in the manuscript.

And the inhibition of PRL were positive.

GCs are Granule cells. I added the full name in the manuscript.

Comments 7: Conclusions

Please edit the conclusions.

-        L304 – L307. “PRL had no significant effect on…” “PRL affected…” It seems that PRL was injected.

Response 7: Thank you for pointing this out. We agree with this comment. Therefore, we have changed some information in the manuscript.

Comments 8: Figures

-        Figure 2. Concentration of what? This applied to all the figures, except A.

-        Figure 3. Indicate the n per figure.

-        Figure 4 - 6. Title in X axis is missing.

Response 8: Thank you for pointing this out. We agree with this comment. Therefore, we have changed figures in the manuscript.

Reviewer 2 Report

Comments and Suggestions for Authors

This study aims to explore the effect of prolactin on secretion of reproductive hormones and gene expression of ewes during the estrus cycle.. The comments on the study are shown below. Unfortunately, this study lacks novelty and needs serious revision and completion of missed materials and results. It cannot be recommended for publication in its present form.

The timing of pro-estrus, estrus, met-estrus, and diestrus seems difficult to be controlled as mentioned in the materials and methods.

Line 23: with good estrus…change with overt signs of estrus

Line 44: revise the sentence states that one dominant follicle is selected in sheep for ovulation, it is not true for all sheep breeds as some breeds are polytocus.

Introduction: the introduction is so generalized and does not reflect adequately the expected role of prolactin and its relationship with other reproductive hormones and estrous cycle and behaviour. Sure, ther are previous studies used the same protocol applied in this study to detect the role of prolactin. Some of these studies should be shown to make the introduction more specific and profound.

Line 75: Sixty two healthy ewes (body weight=52.98±0.96kg) were selected form each groups. ..Do you mean for both groups or each group? Revise because it contracts what mentioned later in lins 79 and 86

Line 79: What is the breed name of ewes used in this study? What is the meaning of post-weaning ewes??? Please, specify clearly the physiological status of these animals, age, breeding season (out or in breeding season)

Line 84084: use eCG instead of PMSG

Line 90: on the day of the burial of the embolism in the same estrus treatment… what is the day of burial ??? do you mean day of sponge insertion?.. please, modulate

Line 86: 31 sheep with good estrus were selected for next. The experiment was lasted over 45 days…the sentence needs re-editing..do not start with number (31) and next to what???

For hormones: all information regarding the source, specificity, sensitivity, inter/intra assay covariance should be shown.

Melatonin is mentioned of analyzed hormones, while no shown results available.

Line 109: add reference for follicle classification

Other comments:

Use the American or British style in Editing.. for example estrus or oestrus

Differentiate between estrous (adjective) and estrus (noun) and use them properly

Comments on the Quality of English Language

The article needs moderate revision for adjusting quality of English and the use of scientific expressions in the field.

Author Response

Comments 1: The timing of pro-estrus, estrus, met-estrus, and diestrus seems difficult to be controlled as mentioned in the materials and methods.

Response 1: Thank you for pointing this out. We have added the corresponding information.

We added “Ewes in proestrus were determined according to records of estrus cycles: the proestrus stage is 1 day before the estrus, the estrus stage is 1 day after the proestrus; the metes-trus stage is 2 days after estrus; and the diestrus stage is 7 days before the estrus. The first and second estrus times were accurately recorded, and the number of estrus cycle days was counted by subtracting the first estrus date.  “ in experiment, see the red section of the article for details.

Comments 2: Line 23: with good estrus…change with overt signs of estrus

Response 2: Thank you for pointing this out. We have revised the description.

After estrus synchronization, thirty-one ewes with overt signs of estrus were selected from each group.

And we updated text in the manuscript and highlighted it.

Comments 3: Line 44: revise the sentence states that one dominant follicle is selected in sheep for ovulation, it is not true for all sheep breeds as some breeds are polytocus.

Response 3:

Thank you for pointing this out. We agree with this comment. We have revised this part of the manuscript. We removed the "a single" from the manuscript.

Comments 4:

Introduction: the introduction is so generalized and does not reflect adequately the expected role of prolactin and its relationship with other reproductive hormones and estrous cycle and behaviour. Sure, ther are previous studies used the same protocol applied in this study to detect the role of prolactin. Some of these studies should be shown to make the introduction more specific and profound.

Response 4:

Thank you for pointing this out. We agree with this comment. Therefore, we have changed some information in the manuscript.

We have added some content to the introduction, which can be found in the manuscript.

Comments 5: Line 75: Sixty two healthy ewes (body weight=52.98±0.96kg) were selected form each groups. ..Do you mean for both groups or each group? Revise because it contracts what mentioned later in lins 79 and 86

Response 5: Thank you very much. We agree with this comment. Therefore, we have changed some information in the manuscript and Figure1.

Eighty healthy non-pregnant ewes (Hu sheeps, 2-3 years, body weight=52.98±0.96kg ) were selected, and randomly divided into a control group (C)(n=40)and a treatment group (T)(n=40). After estrus synchronization, thirty-one ewes with overt signs of estrus were selected randomly from each of the C group and the T group. See the red section of the article for details.

Comments 6: Line 79: What is the breed name of ewes used in this study? What is the meaning of post-weaning ewes??? Please, specify clearly the physiological status of these animals, age, breeding season (out or in breeding season).

Response 6:

Thank you for pointing this out. We agree with this comment. Therefore, we have changed some information in the manuscript.

The breed name and physiological status of ewes were added in the 2.2.1 Sheep estrus Synchronization Treatment of ewes. The ewes we choosed were those that have ended their lactation period. And the description of post-weaning ewe were  changed to non-pregnant ewes. See the red section of the article for details.

Comments 7: Line 84084: use eCG instead of PMSG

Response 7: Thank you for pointing this out. We agree with this comment. Therefore, we have changed the PMSG.

Comments 8: Line 90: on the day of the burial of the embolism in the same estrus treatment… what is the day of burial ??? do you mean day of sponge insertion?.. please, modulate

Response 8: Thank you for pointing this out. We agree with this comment. Therefore, we have changed figures in the manuscript. There are loopholes in our description of feeding BCR and insertion of implanted progesterone sponge plug.

The description of the trial design has been changed in 2.2.2. See the red section of the article for details.

Comments 9: Line 86: 31 sheep with good estrus were selected for next. The experiment was lasted over 45 days…the sentence needs re-editing..do not start with number (31) and next to what???

Response 9: Thank you for pointing this out. We agree with this comment. Therefore, we have changed figures in the manuscript. We changed the sentence with “After estrus synchronization, thirty-one ewes with overt signs of estrus were selected randomly from each groups“.

Comments 10: For hormones: all information regarding the source, specificity, sensitivity, inter/intra assay covariance should be shown.

Response 10: Thank you for pointing this out. We agree with this comment. Therefore, we have changed figures in the manuscript. We used commercial sheep ELISA kits.

“Following the manufacturer’s instructions, commercial sheep enzyme-linked immunosorbent assay (ELISA) kits from Nanjing Jiancheng Bio, Nanjing, China was used to determine the serum concentrations of PRL (H095-1-2, Sensitivity > 0.1 ng/mL), FSH (H101-1-2, Sensitivity > 0.1 mIU/mL), E2 (H102-1-2, Sensitivity > 0.1 ng/L), LH (H206-1-2, Sensitivity > 0.1 mIU/mL), Progester-one (P4, H089-1-1, Sensitivity > 0.1 ng/mL), and gonadotropin-releasing hormone (GnRH, H297, Sensitivity > 0.1 ng/L). The absorbance (OD) of each well was measured at 450 nm and a standard curve was generated. According to the standard curve, the serum hormone concentration of the test sheep was calculated. The intraassay CV was 10%.”

Comments 11: Melatonin is mentioned of analyzed hormones, while no shown results available.

Response 11: Thank you for pointing this out. We agree with this comment. It was our negligence that we removed this error from the manuscript. We've removed this section.

Comments 12: Line 109: add reference for follicle classification.

Response 12: Thank you for pointing this out. We agree with this comment. Therefore, we have added reference in the manuscript.

Comments 13: Use the American or British style in Editing.. for example estrus or oestrus.

Response 13: Thank you for pointing this out. We agree with this comment. Therefore, We polished the manuscript.

Comments 14: Differentiate between estrous (adjective) and estrus (noun) and use them properly.

Response 14: Thank you for pointing this out. We agree with this comment. Therefore, We polished the manuscript.

Reviewer 3 Report

Comments and Suggestions for Authors

The present study evaluated the effect of PRL on the secretion of reproductive hormones and gene expressions during different estrus stages to explore the regulatory effects of PRL on the sexual cycle of ewes. The results of the present study demonstrated that inhibition of PRL shows no apparent influence on estrus duration, but significantly changes the secretion of reproductive hormones and the expression of hormone-related genes and apoptosis-related genes, indicating a regulatory role in follicle development and ovulation in ewes. This study provides new knowledge of PRL on ewe estrus cycle. There are several points regarding the contents of this study I would like to share my opinions with the authors.

General comments:

This study showed a good presentation of the results. In relation to the research design, it is confusing how many animals were used in this study. It is suggested to state clearly how to select the animals and the final number of animals used in each group. In this study, the authors measured hormones’ level and their gene expressions as well. As the hormones’ levels were significant, is it necessary to test their gene expression? In addition, why did the authors choose to measure apoptosis-related gene expressions? What is the connection between apoptosis-related genes with reproductive hormone levels? Concerning the effect of PRL on ewe estrus, it would be better to present results of estrus behavior of ewes due to inhibition of PRL.

Specific comments:

Q1. In the Introduction section, there is lack of information on those parameters measured in this study, especially those genes. And the progress on the study of those parameter in ewe estrus cycle should be stated. It is important to point out why these parameters were chosen and the link between them, in order to give a clear idea of the purpose of this study.

Q2. Line 75, the number of animals here is different from that in the abstract. Please align all the information and make it clear.

Q3. Line 83, 16,00 should be 16:00, right? Line 86, the 31ewes were selected for the next experiment. The authors should state clearly if it was the total number of ewes used for the next experiment, and how they were divided into two groups? So is in Line 89, the number of ewes used is not clear and the information of control group is missing. I strongly suggest that the authors rewrite the experimental design to make sure that it is clear and easy to understand.

Q4. Line 104, why only 6 ewes were selected for sampling? How many replicates were made for test? I suggest to state the sample source (ovary) for measurements of gene expression in Abstract and experimental design.

Q5. In figure 2, it can be seen that there was a great deviation of values of hormone levels. It comes to the same question: how many replicates were performed? Besides, how to explain the increase of E2 levels while FSH and LH showed no apparent difference? In addition, the concentration of each hormone should be accompanied by their unit.

Q6. For Figure 3, how to explain the different effects of PRL inhibition on estrus? The PRL inhibit or promote follicle development?

Q7. In Discussion section, it is suggested to combine all the parameters together to discuss the effect of PRL inhibition on follicle development and ovulation, thus influencing estrus cycle.

Q8. Unfortunately, the conclusion is not concise and precise as the authors did not connect all the information.

Comments on the Quality of English Language

There are many small errors in grammar. Please check them carefully and make corrections.

Author Response

Comments 1: This study showed a good presentation of the results. In relation to the research design, it is confusing how many animals were used in this study. It is suggested to state clearly how to select the animals and the final number of animals used in each group. In this study, the authors measured hormones’ level and their gene expressions as well. As the hormones’ levels were significant, is it necessary to test their gene expression? In addition, why did the authors choose to measure apoptosis-related gene expressions? What is the connection between apoptosis-related genes with reproductive hormone levels? Concerning the effect of PRL on ewe estrus, it would be better to present results of estrus behavior of ewes due to inhibition of PRL.

Response 1: Thank you for pointing this out. We have added the corresponding information. We have revised the part of the description for experimental ewes, and updated the experimental design and collection protocol in figure1, see the red section of the article for details.

Hormones control developmental and physiological processes, often by regulating the expression of multiple genes simultaneously or sequentially. Hormonal signals can guide the addition and removal of epigenetic marks, steering gene expression. So we test their genes.

Apoptosis is involved in the process of atresia. The occurrence of atresia is accompanied by changes in reproductive hormones.

And there is no difference in estrus behavior for PRL inhibition, so we do not discuss it in this article.

Comments 2: In the Introduction section, there is lack of information on those parameters measured in this study, especially those genes. And the progress on the study of those parameter in ewe estrus cycle should be stated. It is important to point out why these parameters were chosen and the link between them, in order to give a clear idea of the purpose of this study.

Response 2: Thank you for pointing this out. We have added the corresponding information on the reproductive hormones. The tested genes included three parts, the first part is hormone-related receptor genes, the second part is steroid synthesis-related genes, and the third part is apoptosis-related genes. Cycles, prolactin, are described in our introduction, but there is no mention of the genes we measured. Regulation of gene expression is mediated by an array of intracellular receptors. Upon binding to hormones, the receptors interact with specific hormone response elements located in the promoters of numerous genes. So we added the information on hormones.

Comments 3: Line 75, the number of animals here is different from that in the abstract. Please align all the information and make it clear.

Response 3: Thank you for pointing this out. We agree with this comment. Therefore, we have changed some information in the manuscript in Experiment and update the figure1.

Comments 4: Line 83, 16,00 should be 16:00, right? Line 86, the 31ewes were selected for the next experiment. The authors should state clearly if it was the total number of ewes used for the next experiment, and how they were divided into two groups? So is in Line 89, the number of ewes used is not clear and the information of control group is missing. I strongly suggest that the authors rewrite the experimental design to make sure that it is clear and easy to understand.

Response 4: Thank you for pointing this out. We agree with this comment. Therefore, we have changed some information in the manuscript in Experiment and update the figure1.

Eighty healthy post-weaning ewes (Hu sheeps, 2-3 years, body weight=52.98±0.96kg ) were selected, and randomly divided into a control group (C)(n=40)and a treatment group (T)(n=40). After estrus synchronization, thirty-one ewes with estrus response were selected randomly from C group and T group. In proestrus (I), estrus (II), metestrus (III), and diestrus(IV), six ewes were randomly selected to collect blood and ovarian tissue per stage in both C group and T group. See the red section of the article for details.

Comments 5: Line 104, why only 6 ewes were selected for sampling? How many replicates were made for test? I suggest to state the sample source (ovary) for measurements of gene expression in Abstract and experimental design.

Response 5: Thank you for pointing this out. We agree with this comment. In proestrus (I), estrus (II), metestrus (III), and diestrus(IV), six ewes were selected to collect blood and ovarian tissue per stage in both C group and T group. The replicates were six for per stage in both C group and T group. Sheeps after the ovarian collection were no longer involved in the experiment. See the red section of the article for details.

Comments 6: In figure 2, it can be seen that there was a great deviation of values of hormone levels. It comes to the same question: how many replicates were performed? Besides, how to explain the increase of E2 levels while FSH and LH showed no apparent difference? In addition, the concentration of each hormone should be accompanied by their unit.

Response 6: Thank you for pointing this out. We agree with this comment. The replicates of hormone levels were six. An increase in E2 levels was accompanied by a significant increase in FSH during the proestrus period, while there was no significant difference in LH. We found that the concentration trends of LH and FSH were the same as those of E2. When PRL was combined with bromocriptine, GnRH failed to stimulate LH release at all doses tested (P < 0.01) {Gregory, 2004 #214}. And we added the unit of each hormone concentration in the figure2.

Comments 7: For Figure 3, how to explain the different effects of PRL inhibition on estrus? The PRL inhibit or promote follicle development?

Response 7: Thank you for pointing this out. We agree with this comment. We found that inhibition of PRL significantly increased the number of large and small follicles, so we thought inhibition of PRL promote follicle development.

Comments 8: In Discussion section, it is suggested to combine all the parameters together to discuss the effect of PRL inhibition on follicle development and ovulation, thus influencing estrus cycle.

Response 8: Thank you for pointing this out. Our current discussion is divided into three parts: PRL inhibition in the estrous cycle on the number of follicles, reproductive hormone levels, expression of receptor genes and steroid synthesis genes. We think that the role of PRL inhibition in the cycle is more clear in three parts for discussions.

Comments 9: Unfortunately, the conclusion is not concise and precise as the authors did not connect all the information.

Response 9:

Thank you for pointing this out. We agree with this comment. And we changed the conclusion. See the red section of the article for details.

Round 2

Reviewer 1 Report

Comments and Suggestions for Authors

Manuscript has been improved. It can be accepted.

Reviewer 2 Report

Comments and Suggestions for Authors

All of my comments were adequately addressed .

Comments on the Quality of English Language

Minor revision is required to improve English quality

Reviewer 3 Report

Comments and Suggestions for Authors

Generally speaking, the authors have corresponed approriately to my questioned concerning the content of this study. I have no further comments.